# How Raising Tobacco Prices Affects the Decision to Start and Quit Smoking: Evidence from Argentina

**DOI:** 10.3390/ijerph16193622

**Published:** 2019-09-27

**Authors:** Martin Gonzalez-Rozada, Giselle Montamat

**Affiliations:** 1Departamento de Economía, Universidad Torcuato Di Tella, Av. Figueroa Alcorta 7350, C1428BCW Buenos Aires, Argentina; 2Department of Economics, Harvard University, Cambridge, MA 02138, USA

**Keywords:** smoking onset, quitting smoking, split-population model, cigarette prices and taxes, survival analysis

## Abstract

We used a two-part model for the estimation of the price elasticity of participation and consumption of cigarettes by the duration of the smoking habit and a continuous-time split-population model for the estimation of prevalence and duration of smoking onset and smoking addiction, allowing for covariates in the participation component of the model. Results: We computed the total price elasticity of consumption of cigarettes by quartiles of addiction and found that for the people located in the lowest quartile of addiction the total price elasticity is around −0.51; while for those located in the highest quartile of addiction this figure is only −0.19. Then, a 10% increase in cigarette prices, via taxes, reduces the consumption of those in the early stages of the addiction by 5% and for those with a longer history of addiction by only 1.9%. Estimating the continuous-time split-population model we found that, at the mean starting age of 15 years, an increase of 10% in real cigarette prices is expected to delay smoking onset by almost two and a half years. On the other hand, the same policy is less effective to reduce the duration of the habit because there is no meaningful relationship between the duration of the smoking habit and the real price of cigarettes.The policy of raising cigarette excise taxes, to increment prices, seems to be more effective to delay smoking onset. On the other hand, the same policy is less effective to reduce the duration of the habit. A policy recommendation that emerges from this evidence is that for people with a developed addiction a combination of increasing taxes and other public health policies, like cessation therapies, could prove more effective.

## 1. Introduction

During the past decades, abundant evidence has been presented about the noxious effects that tobacco can have on people’s health. A recent report by the World Health Organization (WHO) on global trends in tobacco smoking (2015), [1], estimates tobacco use to be the cause of death of about six million people across the world each year. Tobacco smoking is the leading cause of preventable premature death in the world. Tobacco use and exposure to tobacco smoke are risk factors for numerous cancers, notably lung cancer. According to the Argentinean National Ministry of Health, tobacco use is the main cause of preventable premature death in Argentina, [2]. Smoking takes over 40,000 lives every year, including 6000 non-smokers who die as a consequence of exposure to tobacco smoke. More than 20 billion pesos (about 2.2 billion dollars) are spent annually to deal with diseases caused by tobacco addiction. This expenditure is about 12% of Argentina’s annual health expenditures [3].

The importance of addressing the tobacco epidemic through control policies is evident. Even delaying the age at which individuals start smoking or accelerating the time of quitting can have substantial health benefits. Research suggests that people who start smoking in their teens and continue for two decades or more will die 20 to 25 years earlier that those who never start [4].

Raising excise taxes on tobacco is one of the measures to control and reduce tobacco demand suggested by the WHO in the context of the Framework Convention on Tobacco Control (FCTC). Incrementing cigarette prices, via taxes, by 50% would save 17 to 45 million persons in Latin America and between 1226 and 3236 in Argentina.

Some studies have addressed the impact that higher prices can have on smoking onset and quitting. Many of these studies use a discrete binary choice framework (Probit, Logit, linear probability models) to model smoking behavior, [5] (see for example Liu 2010 and the references there). These studies consider an indicator-dependent variable for the decision to start or to quit smoking. [6,7,8], among others, analyze the impact of prices on smoking participation and intensity and the decision to quit. In general, results suggest a negative relationship between the price of tobacco products and the decision to initiate smoking. Results regarding quitting do not provide evidence of a straightforward relationship between the increment of prices and quitting; moreover, some authors find that higher prices may not necessarily encourage quitting.

An alternative approach to modeling smoking habits is to use a duration (or survival) analysis, which concentrates not only on whether but on when an event occurs. When analyzing smoking onset (or quitting) it is important to account for the possibility that people never having initiated the addiction will indeed never incur the habit and for smokers who will never quit. A split-population model explicitly accounts for these facts and treats the probability of eventual smoking or quitting as an additional element to be estimated. Using this kind of model, [9] did not find evidence that higher prices would have a significant impact on smoking initiation in the US. [10] found positive effects for the impact of tobacco prices on smoking onset in Great Britain (1920–1984). [11] conducted both a traditional duration analysis and a continuous-time split-population model on data from Vietnam to analyze the effect of taxes on smoking onset. He found evidence of a relatively large and significant effect of prices on the age individuals take on the habit. In the case of Latin American countries, [12] is the only study so far to have undertaken an analysis of the impact that tobacco prices can have on smoking onset. They used data from Argentina and the methods they employed include discrete-time hazard models, a complementary log–log specification and a discrete-time split-population model. Their results suggest a large, negative and statistically significant relationship between the prices of tobacco products and the hazard of smoking; a limitation in their analysis is that they do not allow for explanatory variables in the participation equation of the split-population model.

The aim of this paper is to examine the role of tobacco prices on smoking onset and quitting using data obtained from the Global Adult Tobacco Survey (2012) in Argentina. Our contribution is twofold. First, we estimate cigarettes’ demand price elasticity by duration of the smoking habit using a two-part model. Second, we use a continuous-time split-population model to perform duration analysis. In this last case, our specification for prevalence and duration of smoking onset and quitting allow for covariates in the participation component of the model. No study, to the best of our knowledge, has yet been conducted to measure either the demand price elasticity by addiction levels or the effect of prices on quitting using data from Argentina.

We find that, while negative as expected, the price elasticity is larger in absolute terms for people with a shorter history of addiction than for those that exhibit a longer duration of the habit. In addition, increasing cigarette prices, using taxes, has a larger effect on the starting age of smoking than on the quitting age. This evidence shows that raising cigarette excise taxes, to increment prices, seems to be more effective to delay smoking onset. On the other hand, the same policy is less effective to reduce the duration of the habit. A policy recommendation that emerges from this is that for people with a developed addiction a combination of increasing taxes and other public health policies, like cessation therapies, could prove more effective.

## 2. Methodology

We specify a two-part model, [13,14], in which the cigarette price variable interacts with an addiction variable in this way, we estimate the price elasticity of participation and consumption taking into account the duration of the smoking habit. Then, we use a split-population model, [15], to estimate the impact that an increase in prices can have on the age of starting and quitting smoking.

The two-part model includes two equations, an equation for smoking participation and a second one describing cigarette consumption conditional upon participation. In a first stage, we estimate a Probit model for the probability of smoking against cigarettes’ real price and several control variables. The first equation we estimate is:$$Pr_{i}=\Phi \left(\beta_{0}+\beta_{1}cp_{i}+X_{i}\beta_{3}\right)$$
where $Pr_{i}$ is the predicted probability of smoking cigarettes for individual i and $cp_{i}$ is the natural logarithm of the cigarette price paid by individual i. For individuals that do not smoke, prices are imputed using a random regression imputation. $X_{i}$ is a vector of other control variables including gender, dummies for age categories, for being out of the labor force, for being a housekeeper and for the level of education attained by the individual.

Participation price elasticity is computed using the formula:$$\varepsilon_{i}=\frac{\partial Pr_{i}}{\partial cp_{i}}\times \frac{1}{Pr_{i}}$$

We compute this elasticity at the quartiles of the duration of the addiction and at the sample mean of price and the other control variables.

In a second stage, we consider only the sample of those who smoke and estimate by OLS the regression of consumption (the logarithm of the number of cigarettes smoked by individual i) against the logarithm of price, the duration, the interaction of duration and price and the other control variables specified earlier. The second equation we estimate is then:$$C_{i}=\beta_{0}+\beta_{1}cp_{i}+\beta_{2}cp_{i}\times d_{i}+\beta_{3}d_{i}+X_{i}\beta_{4}+u_{i}$$
where $d_{i}$ is a variable measuring the duration of the habit (computed as the difference between the age at the time of the survey and the staring age of smoking).

The price elasticity of consumption is computed using the formula:$$\eta_{i}^{cp}=\beta_{1}+\beta_{2}d_{i}$$

Both participation and consumption equations are estimated using the sampling weights of the GATS survey. We evaluate this consumption elasticity at quartiles of the duration of the addiction. Total price elasticity is computed as the sum of the participation and consumption elasticities. We have a single record per individual of their starting and quitting age, referred to in the literature as single failure data. When estimating the effect on the smoking onset, duration refers to the time that elapses between the risk age of smoking onset and the age of starting. When estimating the effect on the quitting age we consider the period that runs from the moment the individual starts smoking up to the moment he quits. Individuals can have a complete spell or the spell is right censored. We treat time as a continuous variable. To model duration we use a log–logistic distribution for smoking onset and a generalized Gamma distribution for quitting. We also estimate the split-population model by gender.

We model participation with a Probit. The dependent variable is a dummy that indicates whether the individual “fails” (i.e.; starts smoking or quits) or not.

The main idea behind the use of a split-population model is to account for the fact that not all individuals that have an incomplete spell will eventually “fail” (i.e.; start or quit smoking). The duration process applies then only to those individuals who are predicted to eventually “fail”. The likelihood of each observation is weighted with the probability that the individual will ever start (quit) smoking. Formally expressed, the log-likelihood function to be maximized is:(1)$$\ln\left(L\right)=\sum w_{i}\{c_{i}\ln[F\left(\alpha^{\prime }z_{i}\right)f(t/s=1,x_{i}\left(t\right)]+\left(1-c_{i}\right)\ln\;[1-F\left(\alpha^{\prime }z_{i}\right)+F\left(\alpha^{\prime }z_{i}\right)S(t/s=1,\;x_{i}\left(t\right)]\}$$
where *c* is a dummy variable adopting the value 1 if the individual ever smoked (quit) and 0 otherwise. *s* is another dummy variable equal to 1 if the individual will eventually start smoking (will eventually quit) and 0 if he/she never does. *F* is the probability of starting (quitting) which we model using a Probit specification. *z* are time-invariant covariates, *f* refers to the chosen conditional density function to model duration, *S* is the respective survival function and *w* is a survey weight. We treat the price of cigarettes as a time-varying covariate (the length of the time series for the price of cigarettes is 16 years, from 1996 to 2012). All time-invariant covariates are used both in the specification for duration and participation. We are not assuming the probability of ever smoking or quitting as an additional parameter to be estimated. Instead, we consider a Probit model, allowing for the introduction of explanatory variables; something that, to the best of our knowledge, has not been done in previous studies using data from Argentina. Estimations of both the Probit model and the model of duration are done using the sampling weights of the survey. We carried out the estimation in two steps. In the first step, we estimated the probability of starting (quitting) using a Probit model. In the second step, we introduced this estimated probability into Equation (1) and maximized the likelihood.

Since information in the survey is in annual terms, we randomly assigned the month at which the individual starts (or quits) smoking in the reported year of smoking onset (quitting).

## 3. Findings

### Data and Descriptive Analysis

The Global Adult Tobacco Survey (GATS) is a nationally representative household survey of adults 15 years of age and older. The survey systematically monitors adult tobacco use and tracks key tobacco control indicators. In Argentina, it was implemented by the National Institute of Statistics and Censuses (INDEC) in 2012 using a multistage stratified cluster sample design in urban areas. The sample size is 9790 households.

We define a smoker as a person that currently smokes daily or occasionally. A former smoker is a person who has smoked at least 100 cigarettes in his life but currently does not smoke. We classify current and former smokers as “ever smoke”. We assume individuals are first exposed to the risk of starting to smoke at age 11. This assumption seems reasonable based on evidence that suggests that smoking onset, among high school students, is common at age 12 or 13, and even earlier, at 11 (Ministry of Health in Argentina). Almost half the students, who smoke, interviewed in the GATS of 2012, started smoking at age 12 or 13, and 29.1% at 11 or earlier.

We took data on the price of cigarettes starting in January 1996 then we trimmed our sample by excluding all individuals older than 27 in 2012, the date of the survey. Note that the individuals considered are relatively young, particularly with respect to the risk age of smoking onset, which reduces the possibility of recall bias, a usual critique against the use of retrospective data (we perform estimations also for the full sample obtaining similar results).

Based on the individual’s response, two things can happen with smoking onset. The individual can either have a complete spell (i.e.; started smoking before 2012) or an incomplete spell (in which case he can either begin to smoke after 2012 or never start smoking). This smoking onset sample includes 1674 individuals representing (using the sample weights) a population of 8,114,714 persons. For the analysis of the duration of the smoking habit we only consider individuals who began smoking after January 1996, the first date for which we have cigarette prices. Two things can happen: They quit smoking before the date of the GATS survey or they did not. For those who quit smoking, we observe the complete spell. For those individuals that did not quit, we only observe an incomplete spell. In this last case, they can either quit smoking sometime after the date of the survey or never quit. This final sample includes 762 smokers representing a population of 2,799,920 individuals.

The prevalence of individuals who have ever smoked is around 30%. The prevalence for those who currently smoke is around 23%, so the sample includes 7% former smokers. The probability of being a smoker, current or former, is larger for men, 33%, than for women, 27%. In Argentina, 45% of the population with primary education is either a former or current smoker, and this figure is 42% for those with a university education. The prevalence of ever having smoked has a negative relationship with socioeconomic status. The mean starting age of smoking is 15 years old. It is a little bit larger for men, almost 16, than for women, 15 years old. Figure 1 shows the hazard of initiating the habit. Male teenagers have the highest risk of picking up a smoking habit around the age of 19 while for females the highest risk is around 21 years old. The hazard of initiating the addiction increases sharply around the age of 13 for both men and women and falls after the age of 19 for men and 21 for women. This increment in the hazard function is steeper for men than women such that after the age of 15 and until the age of 20 the hazard rate is higher for men than for women. Before the age of 15 and after the age of 20 the hazard rate is higher for women.

Once the individual initiates in the smoking habit the average duration of the addiction is around 26 years. The average duration of the habit is higher for men, 28 years than for women, 24 years. Figure 2 shows the cumulative hazard function of quitting the habit by gender. As reported already in the literature, [10], the figure suggests the hazard function of quitting smoking shows positive duration dependence for both men and women.

## 4. Results

Cigarette price variable is the average retail price of a 20 cigarettes pack published by the National Ministry of Agriculture (MINAGRI), [16]. We transformed the price variable into real terms by deflating it by the Argentinian Consumer Price Index (CPI). Figure 3 depicts the time evolution of cigarettes’ real price from January 1996 to September 2012, the date of the GATS survey. The average real price of a 20 cigarettes pack in the sample is argentine pesos (ARS) 1.50. The maximum price was ARS 2.05 while the minimum real price was ARS 1.22. The figure shows no upward or downward trend for real price over the whole sample.

The tax structure on cigarette consumption in Argentina is very complex. Federal taxes affecting cigarettes are four ad valorem taxes: The additional emergency tax (IAE), the value added tax (VAT), the special tobacco fund (FET) and the internal tax (II). The tax base of each one is different. For example, the tax base of the IAE is the retail price, but the tax base of the II is the retail price minus the VAT, IAE and FET. This implies that changing any tax rate affects the base tax of all others. However, during the period analyzed here there were no changes in tax rates. All real price variation is due to inflation and the industry’s price changes.

Using the two-part model we estimated the total price elasticity of consumption by quartiles of addiction. For the people located in the lowest quartile of addiction (those with less than seven years in the habit), the total price elasticity is around −0.51. For those located in the highest quartile of addiction (more than 48 years in the habit), total price elasticity is only −0.19. This evidence means that a 10% increase in cigarette prices, via taxes, reduces the consumption of those in the early stages of the addiction by 5%. This figure is only 1.9% for those with a longer history of addiction.

This empirical evidence suggests that increasing tobacco taxes is a public policy that seems to be more effective for those people that are somewhere between the onset of smoking and the development of the addiction. On the other hand, for those in more advanced stages of the smoking habit, increasing taxes would be a less effective policy as a means to reduce the use of tobacco.

Table 1 shows the results of estimating a split-population duration model for smoking onset. Columns under the header of “Participation” report the parameter estimation of the prevalence component of the model. Columns under the header of “Duration” report parameter estimation of the duration component of the split-population model. The table shows the estimation for all individuals and the estimation by gender. For all individuals, the coefficient on the log of the price variable is positive in all specifications, suggesting a direct effect of prices on the age of starting smoking. Using the third duration specification estimation, we find that the mean expected survival time in the onset of smoking is around 115 months or about nine and a half years. The implied elasticity of the mean survival time with respect to cigarette prices is 2.46. This evidence suggests that at the mean starting age of 15 years an increase of 10% in prices is expected to delay smoking onset by almost two and a half years. This evidence is an important result from the public policy perspective since it highlights that increasing prices via taxes will delay the initiation of the smoking habit.

Estimations include several controls that could affect the age of starting smoking: Gender, region of residence, year of birth and the spell length. The variable region of residence includes: The northeastern region (NEA), the northwestern region (NOA), the west–center region (Cuyo), the Pampas region and the south region that is the base category in the tables. The variable year of birth would capture the fact that those born earlier could have less information about the tobacco epidemic and, therefore, start smoking earlier. The spell length is included to capture the effect of other covariates at the age of starting smoking not observed at the time of the survey (i.e.; income/wealth, smoking behavior of parents, etc.).

For all individuals, Table 1 reports a positive and statistically significant coefficient, on the year of birth variable, in the duration component of the third specification, supporting the interpretation mentioned above. The same variable has a negative and statistically significant coefficient in the participation equation implying that those born more recently have less probability of smoking. The region of residence does not seem to affect smoking onset or prevalence of smoking, nor does gender.

In the last two specifications, Table 1 reports results for the split-population model by gender. For both men and women, the coefficient on the price variable is positive and statistically significant, suggesting that an increase in prices will delay smoking onset. For males, an increment of 10% in cigarette price induces a delay in smoking onset of about 4% while this figure is 5.7% for women.

We use a generalized Gamma function for the duration of the epidemic to take into account the positive duration dependence of the hazard of quitting the habit. Table 2 reports the results of our estimations for all individuals and by gender. The table has the same structure as Table 1. The second column in each specification reports the estimation of the probability of quitting.

For all individuals, in the first specification, the coefficient on the cigarette price variable is negative, while it is positive in the other two. However, in all specifications, it is not statistically significant, meaning that there is no meaningful effect of prices on the age of quitting.

We included as control variables: Gender, year of birth, region of residence and educational categories. The coefficient associated with the year of birth is negative and statistically significant in the quitting probability equation, suggesting younger people have fewer chances of quitting. Those with secondary education have less probability of quitting when compared to those with primary education. Gender does not seem to affect either the duration of the addiction or the probability of quitting.

When looking at the results by gender we see that the coefficient on the price variable in both specifications is not statistically significant, suggesting there is no meaningful relationship between prices and the reduction of the duration of the epidemic.

Overall, the empirical evidence presented in this section shows that increasing cigarette prices, via taxes, could delay the age of starting smoking but would have no effect on the quitting age.

## 5. Discussion and Conclusion

In this paper, we studied the role of tobacco prices on smoking onset and quitting in Argentina. We estimated a two-part model for the price elasticity of consumption and a continuous-time split-population model for prevalence and duration using data obtained from the GATS (2012) The two main findings were: From the estimation of the two-part model, a 10% increase in cigarette prices, via taxes, reduces the consumption of those in the early stages of the addiction by 5% while this figure is only 1.9% for those with a longer history of addiction; and from the split-population model, at the mean starting age of 15 years, an increase of 10% in real cigarette prices is expected to delay smoking onset by almost two and a half years, while there is no meaningful relationship between the duration of the smoking habit and the real price of cigarettes. Both results suggest that the policy of increasing cigarette excise taxes seems to be more effective to delay smoking onset. On the other hand, the same policy is less effective to reduce the duration of the habit. A policy recommendation that emerges from this evidence is that, for people with a developed addiction, a combination of increasing taxes and other public health policies, like cessation therapies, could prove more effective.

## Figures and Tables

**Figure 1 ijerph-16-03622-f001:**
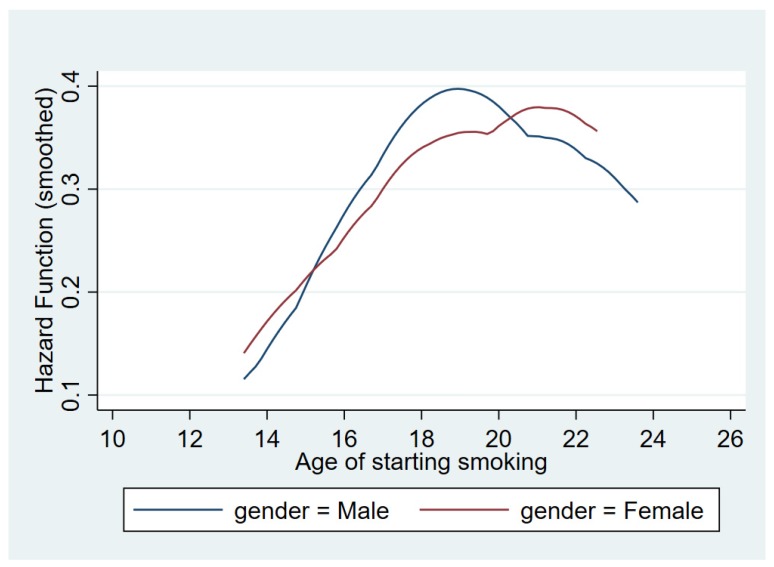
Hazard of initiating a tobacco addiction. Source: Authors’ calculations.

**Figure 2 ijerph-16-03622-f002:**
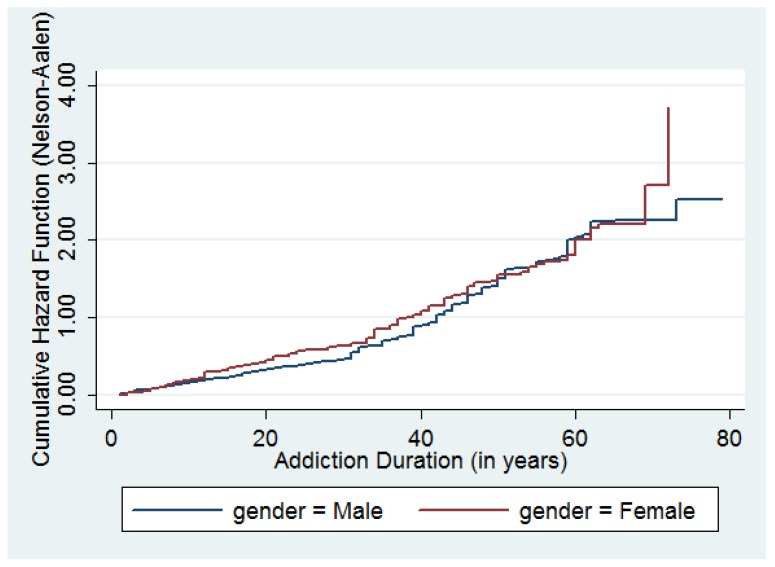
Cumulative hazard of quitting a tobacco addiction. Source: Authors’ calculations.

**Figure 3 ijerph-16-03622-f003:**
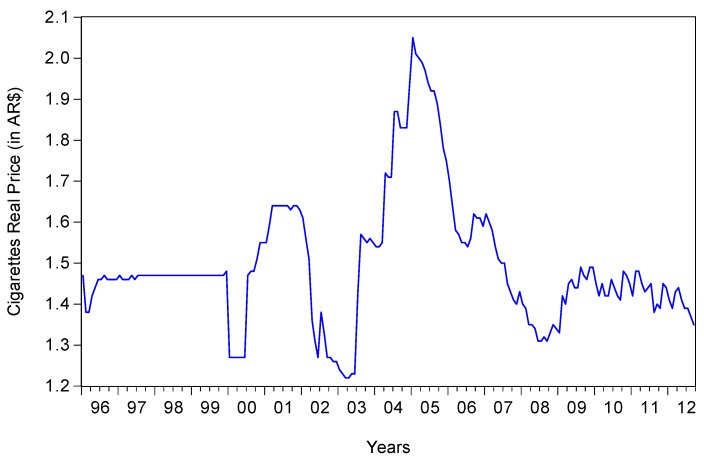
Real price of a pack of 20 cigarettes. Source: Authors’ calculations. CPI April 2008 = 100.

**Table 1 ijerph-16-03622-t001:** Split-population estimations for the onset of smoking.

	All Individuals						Males		Females	
	Duration	Participation	Duration	Participation	Duration	Participation	Duration	Participation	Duration	Participation
Ln(cigarette price)	0.530		0.291		0.496***		0.405**		0.567**	
	(0.785)		(0.219)		(0.150)		(0.162)		(0.271)	
Gender (female = 1)	−0.201	−0.199	−0.088*	−0.149	−0.060	−0.395**				
	(0.175)	(0.018)	(0.050)	(0.178)	(0.043)	(0.189)				
Region										
Cuyo	0.228*	−0.137	0.045*	−0.084	0.049*	−0.122	0.056	0.082	0.039	−0.343
	(0.154)	(0.135)	(0.035)	(0.146)	(0.037)	(0.185)	(0.036)	(0.258)	(0.088)	(0.269)
NEA	−0.016	−0.192	0.001	−0.203	0.001	−0.112	−0.001	0.060	0.043	−0.264
	(0.164)	−0.129	(0.040)	(0.137)	(0.031)	(0.186)	(0.048)	(0.270)	(0.054)	(0.247)
NOA	0.270	−0.097	0.011	0.039	−0.003	0.170	0.02	0.223	−0.036	0.131
	(0.191)	(0.127)	(0.036)	(0.138)	(0.032)	(0.179)	(0.031)	(0.253)	(0.068)	(0.248)
The Pampas	0.040	−0.055	0.020	−0.093	0.010	−0.049	0.042	0.171	−0.015	−0.296
	(0.161)	(0.165)	(0.042)	(0.166)	(0.035)	(0.201)	(0.049)	(0.289)	(0.047)	(0.267)
Birth year					0.023***	−0.690***	0.021***	−0.636***	0.018	−0.772***
					(0.006)	(0.065)	(0.006)	(0.097)	(0.014)	(0.077)
Spell length			0.013***	−0.013***	0.013***	−0.056***	0.013***	−0.053***	0.014***	−0.061***
			(0.000)	(0.002)	(0.000)	(0.006)	(0.000)	(0.008)	(0.001)	(0.007)
Intercept	4.219***	−0.349***	3.161***	0.822***	−42.726***	1379.57***	−40.596***	1270.62***	−34.027	1542.19***
	(0.176)	(0.122)	(0.092)	(0.202)	(12.910)	−129.044	(12.201)	(193.68)	(29.507)	(154.69)
Rho (shape parameter)	0.401***		0.127***		0.112***		0.114***		0.105***	
	(0.039)		(0.017)		(0.015)		(0.025)		(0.014)	

Source: Authors’ estimations. Note: Bootstrap standard errors in parentheses for the duration coefficients and asymptotic standard errors for the participation coefficients. Statistical significance: * 10%, ** 5% and *** 1%.

**Table 2 ijerph-16-03622-t002:** Split-population estimations for the duration of the habit.

	All Individuals						Males		Females	
	Duration	Quitting Prob.	Duration	Quitting Prob.	Duration	Quitting Prob.	Duration	Quitting Prob.	Duration	Quitting Prob.
Ln(cigarette price)	−0.524		0.149		0.153		0.252		−0.139	
	(0.705)		(0.280)		(0.429)		(0.303)		(0.985)	
Gender (female=1)	−0.207	0.257	−0.026	0.268	−0.030	0.241				
	(0.217)	(0.254)	(0.039)	(0.254)	(0.048)	(0.266)				
Region										
Cuyo	−0.195	−0.082	−0.097	−0.082	−0.099	−0.093	−0.053	−0.164	−0.115	−0.046
	(0.245)	(0.293)	(0.079)	(0.289)	(0.096)	(0.289)	(0.075)	(0.377)	(0.160)	(0.445)
NEA	−0.662	−0.234	−0.177**	−0.273	−0.173**	−0.261	−0.179	−0.027	−0.163	−0.145
	(0.218)	(0.254)	(0.075)	(0.256)	(0.076)	(0.246)	(0.174)	(0.340)	(0.108)	(0.377)
NOA	−0.259	−0.190	−0.076	−0.213	−0.071	−0.260	−0.012	−0.608	−0.093	−0.437
	(0.167)	(0.215)	(0.049)	(0.220)	(0.052)	(0.221)	(0.052)	(0.404)	(0.095)	(0.333)
The Pampas	0.006	0.184	−0.055	0.179	−0.055	0.183	−0.012	0.280	−0.056	−0.037
	(0.125)	(0.217)	(0.039)	(0.216)	(0.038)	(0.223)	(0.061)	(0.305)	(0.069)	(0.300)
Educational categories										
Secondary education		−0.715*		−0.755**		−0.781**		0.184		−1.148**
		(0.377)		(0.375)		(0.386)		(0.432)		(0.516)
University education		−0.248		−0.289		−0.302		0.282		−0.375
		(0.397)		(0.398)		(0.414)		(0.477)		(0.532)
Birth year					0.002	−0.041*	0.009	−0.182***	−0.002	−0.013
					(0.004)	(0.023)	(0.006)	(0.028)	(0.004)	(0.016)
Spell length			0.011***	−0.002	0.011***	−0.005*	0.012***	−0.014***	0.011***	−0.006*
			(0.001)	(0.002)	(0.001)	(0.003)	(0.001)	(0.004)	(0.001)	(0.003)
Intercept	4.788***	−0.675**	3.379***	−0.413	0.035	80.834*	−14.252	361.141***	6.786	25.814
	(0.283)	(0.335)	(0.087)	(0.419)	(7.905)	(46.495)	(11.967)	(55.881)	(8.471)	(32.455)
Kappa (shape parameter)	9.251		6.771***		6.764***		6.188***		7.169***	
	(17.064)		(0.348)		(0.338)		(1.414)		(1.087)	
p (shape parameter)	0.559***		2.354***		2.358***		3.097***		2.149***	
	(0.133)		(0.500)		(0.287)		(1.482)		(0.456)	

Source: Authors estimations. Note: Bootstrap standard errors in parentheses for the duration coefficients and asymptotic standard errors for the participation coefficients. Statistical significance: * 10%, ** 5% and *** 1%.

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
