# Peer review of "How Raising Tobacco Prices Affects the Decision to Start and Quit Smoking: Evidence from Argentina"

_ijerph, 2019, doi:10.3390/ijerph16193622_

Round 1

Reviewer 1 Report

I enjoy reading the paper. Few minor comments:

References on line 49 are old. Perhaps a new research can be added. Line 22, " he never do" can be changed to  " he/she never does" Authors created SES index but I could not see its use. Why did not they use it on the analysis?  It is important to indicate CPI year, when estimating real price on Figure 3.

Moderate comments:

Authors need to speak about the taxes on cigarettes. There is no discussion about taxes in the paper and how they evolved (changes)  between 1996 and 2012? Since the main argument is around price and taxes, I suggest authors show and discuss nominal prices on Figure 3. That is because,  it is important to understand the reasons for real price variations. Is it because of the economic crises that inflated inflation, is it because manufacturers did not respond to -any tax changes- during that period?   Likewise, it is important to discuss the income growth during that period.  As the price variable (1996-2012) have been used in the analysis, why did not authors use GDP/capita or disposable income for the whole sample? Did the changes in income (e.g. GDP/capita or disposable income) undermine the changes in price/tax over the years? These are important background information that need to be provided in the document. 

Author Response

Answers to Reviewer #1

References in line 49 were updated. Line 122, sentence was changed. We create the SES variable only to assess the negative relationship between prevalence and SES since the survey does not ask for household income. In the new version of the paper we dropped this part of the text. CPI year was added in the description of Figure 3. We include a description of the cigarette tax structure in Argentina and discussed the evolution of cigarette prices to provide an understanding of real price variations. We did not use alternative measures of income (GDP/capita etc.) because these series in Argentina have quarterly frequency and we have monthly data. We think is better to use more disaggregated data.

Reviewer 2 Report

Dear authors,

Thanks you for the opportunity to review your paper. It is a very interesting paper, with a novel approach and very relevant results. I have enjoyed reading it.

However, I think that some parts of the paper could/should be improved, especially the methodological part and the discussion of the results. Firstly, the paper talks about applying the two-part model, but the methodological section does not explain it clearly step by step (there is a presentation in the appendix, which should rather be incorporated in the main text). Secondly, some parts of the findings section should rather be part of the methodology as they discuss the method. Thirdly, the findings seem to be more about the discussion of results of duration analysis. rather than of the two-part model results. Fourthly, the approach seems to assume that each individual may have only up to 1 failure between 1996 and 2012. What if someone has more than one spell during this period? How is that accounted for? 

Finally, one comments on the findings; the authors conclude that increasing taxes on tobacco does not have an impact on those smokers at the advance stage of addition. However, that may be correct, as this assumes a response to the same increase in price for those less and those more addicted. It may rather be that the more advance stage of addiction requires more significant increase in tax/price for the consumption to respond. That is, that the relationship is not linear.

Author Response

Answers to Reviewer #2

We tried to improve the methodological part of the paper and the discussion of results. First we incorporated the appendix discussing the two-part model in the main text as suggested. We added some explanation of how the equations were estimated and the results in terms of the total elasticities estimated.

We tried to clarify in the discussion of the results what were the findings from the estimation of the two-part model and what were from the split-population model.

In our approach we assume that each smoker has only one failure between 1996 and 2012 as the referee suggest. For those individuals with more than one spell we considered as different individuals with only one spell. For example, if the smoker has two spells, for this observation we consider two individuals with one spell each. We also consider estimations dropping individuals with more than one spell and the results are very similar to those presented in the paper.

Reviewer 3 Report

In the intro please expand on "Some studies have addressed the impact that higher prices can have on smoking onset and 55 quitting. Many of these studies use a discrete binary choice framework (Probit, Logit, linear 56 probability models) to model smoking behavior".

The paper seems to be missing a true discussion of the finds and its implications. The paper simply reports the findings without addressing rational for said findings of its applicability within the political climate.

Author Response

Answers to Reviewer #3

We expanded the description of the models using binary variables to address the impact of prices on smoking onset and quitting as requested.

Due to space limits we did not discuss the applicability within the political climate. However in the final version we included the cigarette tax structure in Argentina which is very complex to give an idea of the difficulty of change taxes. In terms of the applicability, in May 2016 the government decided to increase the rate of one of the taxes, the internal tax, from 60% to 75% until December 2017 and then in that month it pass a Law incorporating a minimum tax (as an specific tax) of AR$ 28 per package of 20 cigarettes. So, there is room change taxes in the country.